# Sex Disparities in the Direct Cost and Management of Stroke: A Population-Based Retrospective Study

**DOI:** 10.3390/healthcare12141369

**Published:** 2024-07-09

**Authors:** Jorgina Lucas-Noll, José L. Clua-Espuny, Misericòrdia Carles-Lavila, Cristina Solà-Adell, Íngrid Roca-Burgueño, Anna Panisello-Tafalla, Ester Gavaldà-Espelta, Lluïsa Queralt-Tomas, Mar Lleixà-Fortuño

**Affiliations:** 1Terres de l’Ebre Healh Region, Catalan Health Service, 43500 Tortosa, Spain; cristina.sola@catsalut.cat (C.S.-A.); iroca@catsalut.cat (Í.R.-B.); 2Department of Primary Care, Institut Català de la Salut, 43500 Tortosa, Spain; apanisellot.ebre.ics@gencat.cat (A.P.-T.); egavalda.ebre.ics@gencat.cat (E.G.-E.); lqueraltt.ebre.ics@gencat.cat (L.Q.-T.); 3Department of Economic and Business, Universitat Rovira i Virgili, 43204 Reus, Spain; misericordia.carles@urv.cat; 4Research Centre on Economics and Sustainability (ECO-SOS), 43204 Reus, Spain; 5Department of Nursing, Universitat Rovira I Virgili, 43500 Tortosa, Spain; mlleixaf@gencat.cat

**Keywords:** cerebrovascular disease, stroke, sex disparities, cost, medical expenses, outcomes

## Abstract

(1) Background: Previous studies have identified disparities in stroke care and outcomes by sex. Therefore, the main objective of this study was to evaluate the average cost of stroke care and the existence of differences in care provision by biological sex. (2) Methods: This observational study adhered to the recommendations of the STROBE statement. The calculation of costs was performed based on the production cost of the service or the rate paid for a set of services, depending on the availability of the corresponding information. (3) Results: A total of 336 patients were included, of which 47.9% were women, with a mean age of 73.3 ± 11.6 years. Women were typically older, had a higher prevalence of hypertension (*p* = 0.005), lower pre-stroke proportion of mRS 0-2 (*p* = 0.014), greater stroke severity (*p* < 0.001), and longer hospital stays (*p* = 0.017), and more were referred to residential services (*p* = 0.001) at 90 days. Women also required higher healthcare costs related to cardiovascular risk factors, transient ischemic strokes, institutionalization, and support needs; in contrast, they necessitated lower healthcare costs when undergoing endovascular therapy and receiving rehabilitation services. The unadjusted averaged cost of stroke care was EUR 22,605.66 (CI95% 20,442.8–24,768.4), being higher in women [*p* = 0.027]. The primary cost concept was hospital treatment (38.8%), followed by the costs associated with dependence and support needs (36.3%). At one year post-stroke, the percentage of women not evaluated for a degree of dependency was lower (*p* = 0.008). (4) Conclusions: The total unadjusted costs averaged EUR 22,605.66 (CI95% EUR 20,442.8–24,768.4), being higher in women compared to men. The primary cost concept was hospital treatment (38.8%), followed by the costs associated with dependence and support needs (36.3%).

## 1. Introduction

Female sex has been identified as an independent predictor of worse in-hospital outcomes among adults with cardiovascular diseases [1,2]. Within this context, stroke outcomes may be influenced by differences associated with biological sex [3]. Therefore, the collection and reporting of more sex-specific data from diverse populations and settings is recognized as a necessity [3,4] toward investigating the increasing demand for sex-specific management guidelines and health policies [5].

Certain neurological conditions, including dementia, multiple sclerosis, migraine, and stroke, disproportionately affect women [3,6]. Stroke, in particular, stands as a leading cause of death and substantial long-term disability worldwide, and, compared to men, women experience increased disability, a poorer quality of life, and a higher likelihood of institutionalization following a stroke. These sex differences in outcomes persist for up to five years post-stroke [7], and it is important to note that poor functional outcomes are linked with higher healthcare costs [8], yet the sex differences in stroke-related healthcare costs remain poorly understood. Women face unique risk factors for cerebrovascular disease, and some traditional stroke risk factors exert a stronger influence in women. Furthermore, disparities persist in research representation, as women continue to be understudied, under recognized, underdiagnosed, and undertreated [4,9,10].

The study area, the Health Region of Terres de l’Ebre, presents an interesting case from a health policy standpoint, given the concurrent presence of factors such as a long distance from the thrombectomy-capable referral center (accessibility to advanced technology), the highest aging index (older stroke profile), and a lower average income. These three factors have been identified within the study area as contributors to health determinant-based inequities compared to the Catalonia region [11,12,13]. Additionally, previous research has highlighted disparities in the management and outcomes of stroke patients between women and men [14]. This suggests a multifaceted landscape where the stroke assistance model, socioeconomic factors, and geographical challenges may intersect with differential costs and outcomes related to gender-specific health disparities, thus warranting targeted policy interventions for improved healthcare equity. In all cases, in accordance with the recommendations from the European Stroke Association, both treatments should be administered at the earliest possible time following hospital admission [15,16,17,18]. In the scenario of patients with acute ischemic stroke resulting from large vessel occlusion and eligible for immediate treatment, potential cost disparities arising from variations in medical assistance times and access to reperfusion treatment based on biological sex remain uncertain. This uncertainty persists despite women exhibiting poorer prognoses, primarily due to factors such as older age and more severe stroke compared to men.

Gaining a better understanding of sex differences in stroke has the potential to lead to earlier and more reliable diagnosis and effective prevention, treatment, and disease management, which together will reduce the impact of these conditions on individuals, families, societies, and economies [6,19]. The main objective of this study was to evaluate the average cost of stroke care and the existence of sex differences in the cost of stroke-related procedures and care complexity, according to previously published frameworks [20].

## 2. Materials and Methods

### 2.1. Study Design

This study was a multicenter, retrospective, observational, and ecological investigation. The International Classification of Diseases (ICD-10) codes were utilized for categorization. The study was completed within the Health Region of Terres de l’Ebre (RSTE), situated in Catalonia, Spain (Appendix A), covering the period from 1 June 2018 to 31 May 2019. The study design adhered to the recommendations of the STROBE Statement.

### 2.2. Territory Scope

According to census data, the RSTE includes a total population of 178,112 individuals, with women comprising 49.6% of the population. Notably, Terres de l’Ebre exhibits a higher aging index (159.5) compared to Catalonia (131.3) and Spain (118.4). This demographic characteristic is particularly pertinent to the study, as the majority of cases involved elderly individuals.

The RSTE comprises four regions, encompassing a total of 11 primary care teams (EAPs), all managed by the Catalan Department of Health. Remarkably, 98.2% of the census population possesses an active medical record within at least one of the EAPs or reference hospitals within the territory. This widespread availability of digitized medical records facilitates continuous monitoring from any public health center across Catalonia.

The Catalan Stroke network presently encompasses 6 comprehensive stroke centers, 8 primary stroke centers, and 15 tele-stroke centers (Appendix A). Emergency medical services, under a single centralized coordination, serve as the primary activators of the stroke code. However, activating the stroke code protocol is also possible for emergency departments in all stroke and non-stroke centers across the Catalonia region, as well as for patients experiencing acute stroke while hospitalized for other conditions.

The patient transport system ensures the urgent and prioritized transfer of suspected acute stroke patients to the nearest stroke center (Appendix A), with prenotification via phone call to the receiving center. In the RSTE, the Stroke Code model involves transportation to a local stroke hospital (drip-and-ship), where treatment with intravenous alteplase (IVT) is available but not endovascular therapy (EVT). If EVT is necessary, the case must be transferred to an EVT-capable referral center, as defined by the Catalan Stroke network.

### 2.3. Participants: Exclusion and Inclusion Criteria

For this study, we included data from 336 patients who were registered as having large vessel occlusion between June 2018 and May 2019 in the electronic databases of the Catalan Department of Health (RSTE). These patients were identified based on neuroimaging (Computed Tomography Scan) or Magnetic Resonance Imaging (MRI), stroke code activation (SCAct) in the Registry Codi Ictus Catalunya (CICat) [21], and eligibility for both intravenous alteplase (IVT) and/or endovascular therapy (EVT) treatments.

According to European Stroke Organisation (ESO) guidelines, patients with acute ischemic stroke (AIS) presenting within 4.5 h after symptom onset may be considered for intravenous thrombolysis and/or mechanical thrombectomy based on specific demographic criteria: age ≥ 18 years, National Institutes of Health Stroke Scale (NIHSS) score ≥ 6, and a baseline modified Rankin Scale (mRS) score of 0–2 [15,16].

All included cases should have an active medical history in any of the healthcare centers with information accessible through the shared medical record (HC3), assignment to any of the primary care teams, and residence in the RSTE territory.

### 2.4. Dataset

Data were sourced from several publicly accessible datasets to obtain demographic, clinical, care, treatment, hospitalization, residual disability, and associated cost data at different stages of evaluation: pre-stroke, stroke episode, post-stroke, and dependence status. The calculated costs, both direct and indirect, were obtained either from the production cost derived from analytical accounting or from the rate paid by the healthcare system to the provider for a specific service or set of services. The choice depends on the availability of the corresponding information:

#### 2.4.1. Health Plan of the Terres de l’Ebre Healthcare Region 2021–2025 [21]

This is a strategic public document and outlines the goals, priorities, and actions that will guide healthcare services in the Terres de l’Ebre region of Catalonia, Spain, from 2021 to 2025.

#### 2.4.2. Statistics Institute of Catalonia [11,12]

This public dataset includes information on gross household income per capita relative to the Catalonia average (100%), population density per square kilometer, and aging index. Additionally, it provides the territorial socioeconomic index (IST), which comprises information on the employment status, educational level, immigration, and income of all individuals residing in each territorial unit relative to the Catalonia average (100%). This index summarizes several socioeconomic characteristics of the population into a single value.

#### 2.4.3. Shared Clinical Record of Catalonia (HC3)

This is led by the Department of Health of the Government of Catalonia. The HC3 aggregates the set of documents containing data, information, and clinical assessments about the situation and evolution of a patient throughout their care process. The HC3 allows for the interconnection between various systems and initiatives developed (electronic prescription services, medical image digitization systems, and telemedicine systems), even when they are promoted by different providers. These databases will allow the inclusion of both outpatient populations in any type of coded care in primary care, including those receiving home care as well as individuals who are institutionalized in centers covered by the referring EAPs.

#### 2.4.4. Registry Codi Ictus Catalonia (CICat) [22]

This is a government-mandated, hospital-based dataset that includes all incidences of stroke treated with an activated stroke code [22], within the Catalonia Stroke Code system [https://aquas.gencat.cat/ca/fem/intelligencia-analitica/registre-cicat/ (accessed on 10 April 2024)]. The data from the study region were compared to the average data from Catalonia whenever available.

#### 2.4.5. Planifi.cat Project

This is an active patient scheduling tool for those with chronic conditions such as cardiovascular diseases or disabilities. An algorithm designed for this purpose suggests the appointments and tests for monitoring according to each patient’s specific pathologies. In this study, it is used to define the follow-up protocol and the number of visits for patients diagnosed with stroke recorded in their medical history (https://si9sapics.wordpress.com/2023/02/22/planifi-cat-programacio-proactiva-de-pacients-cronics/ (accessed on 10 April 2024)).

#### 2.4.6. Department of Social Action and Citizenship

Government of Catalonia. According to a person’s need for assistance in performing the basic activities of daily living, three degrees of dependency are established based on the Dependency Assessment Scale (BVD): moderate dependency (Grade I), severe dependency (Grade II), and high dependency (Grade III). Official recognition of an individual’s dependency situation under one of the established grades is a basic requirement to access economic benefits and social services, as decreed by the Dependency Law [23].

#### 2.4.7. Official Gazette of the Government of Catalonia Order

SLT/71/2020, 2 June 2020, (https://portaljuridic.gencat.cat/ca/document-del-pjur/?documentId=875147 (accessed on 3 April 2024)): This publication details the costs of different healthcare concepts and procedures. It reports the dates of the unit costs and the rate paid by the healthcare system to the provider for a specific service or set of services, plus the currency and year of conversion. When a procedure does not exist, the average costs from previously published references [20] are used instead.

### 2.5. Variables

Primary outcomes: The average total cost of stroke-related procedures and care complexity by sex.

The variables were grouped into the following categories:Demographic information and IST.Pre-stroke clinical: Registered cardiovascular risk factors; functional status by modified Rankin Scale (mRs) and/or Barthel score; adjusted morbidity groups; stroke code activation; type of transportation pre-hospital; and in-hospital mortality.Stroke episode: transportation to a local stroke center (drip-and-ship) and/or a thrombectomy-capable (mothership model) referral center; stroke type via neuroimaging using Computed Tomography Scan (CT Scan) and Magnetic Resonance Image (MRI); reperfusion therapy (IVT/EVT); and standard hospital resources.Post-stroke (90 days): Long stay in hospital; functional outcomes using the mRs scale and/or Barthel scores for stroke severity assessed at hospital discharge; and outpatient follow-up costs post-hospital discharge. A good functional outcome was defined as an mRS scale ≤ 2 at 90 days.Dependence post-stroke (1 year): This is the number of cases of dependency registered during the period divided by the population or time at risk. The unit for person time in this study was person-years, according to the Dependency Law [23,24]. The costs related to sick leave due to stroke were not calculated, given that the stroke incidence was 0.357 per 10,000 person-years before the age of 65, compared to 21.1 per 10,000 person-years for those aged 65 and older [23]. The formal costs associated with the severity level recognized by the Department of Social Action and Citizenship were estimated according to the scales defined in the Dependency Law [23]. Informal costs were based on known average costs and the percentage of patients with caregivers.Stroke cost and follow-up: The model of payment in the public health system is structured around a set of activities and procedures conducted on a patient throughout their continuous hospitalization until discharge or transfer to another facility. Details regarding costs, rates, and the interpretation of variables can be consulted in the Official Gazette of the Government of Catalonia, Order SLT/71/2020, dated 2 June 2020 (https://portaljuridic.gencat.cat/ca/document-del-pjur/?documentId=875147 (accessed on 5 April 2024)).

Thus, distinct payment sets have been defined according to the level of complexity of the hospital: Code stroke activation falls under the comprehensive expenses of emergency service care, while in-hospital stroke treatment incorporates baseline neuroimaging, medical interventions, intravenous thrombolysis, total bed day costs, and in-hospital rehabilitation as standard care. Given the “drip-and-ship” model in the RSTE territory, the patient is evaluated and may be treated with IVT at a community hospital and/or be transferred to a comprehensive stroke center for more advanced interventions such as EVT. Outpatient follow-up costs post-hospital discharge include visits to hospital services, home-based or rehabilitation center rehabilitation, and associated transportation expenses within the initial year post-episode. Primary care visits subsequent to the stroke episode are categorized as secondary prevention costs, according to the Planifi.cat project. To determine the average costs for informal (family or external caregiver) and formal (Dependence Law) care, previously published [20,24] findings in the region have been referenced, excluding individuals who experienced a Transient Ischemic Attack (TIA), mortality within 90 days, and/or institutionalization.

The datasets generated, used, and/or analyzed during the current study are public and/or available from the corresponding author on reasonable request. The STROBE checklist is added to the Appendix A.

### 2.6. Statistical

Descriptive analyses were conducted to describe the study population. Continuous and ordinal variables were characterized by either a mean or a standard deviation (SD). Categorical variables were presented as counts and percentages. Group comparisons based on biological sex utilized the χ^2^ test or the Fisher exact test when necessary for categorical variables. Continuous variables were compared between groups employing a *t*-test, Mann–Whitney test, or ANOVA. The Cox regression curve has been used to calculate hazard ratios. All statistical analyses were performed as two-tailed tests with a confidence interval of 95%, considering *p*-values less than 0.05 as indicative of significance. The analyses were concentrated on the subset of the population that underwent stroke code activation registered on CICat. No imputation methods were employed to substitute missing data. The IST cannot be disaggregated below the level of municipal units for confidentiality reasons and was just used as a cumulative indicator of the territory.

## 3. Results

### 3.1. Clinical Results

A total of 336 patients with stroke code activations were included, with a mean age of 73.3 ± 11.6 years. Women (47.9%) tended to be older, more commonly exhibited previous functional dependence, and presented with higher stroke severity and more severe dependence post-stroke. However, there were no significant differences in terms of stroke activation distribution, clinical characteristics, or stroke treatment (Table 1).

While women were older than men, the difference was not statistically significant (*p* = 0.069). Women had a significantly higher prevalence of arterial hypertension diagnosis (*p* = 0.005), a lower pre-stroke proportion of mRs between 0 and 2 (*p* = 0.014), and lower Barthel scores (*p* = 0.013). They also had a higher usage of conventional ambulances (*p* = 0.022), greater stroke severity (*p* < 0.001), and longer hospital stays (*p* = 0.017), and were more frequently referred to nursing or residential services (*p* = 0.001), which was associated with an increased need for support (*p* = 0.004) at ninety days and one year post-stroke.

TIA episodes (NIHSS ≤ 4) occur among women at a significantly younger age than ischemic episodes (71.4 ± 11.5 vs. 73.3 ± 11.4, *p* = 0.001).

The odds ratio (HR) of receiving IVT for patients arriving at the hospital <1 h after symptom onset was 2.7 times higher compared to those arriving between 1 and 2 h and 3.2 times higher compared to those arriving between 2 and 3 h after symptom onset. No statistically significant differences in functional outcomes (*p* = 0.41) were observed between those who received fibrinolysis and those who did not, but in individuals ≤ 80 years, the use of IVT was associated with higher survival at 1 year in women (HR 0.98, CI95% 0.99–0.97; *p* < 0.001) compared to men (HR 0.77, CI95% 0.65–0.89). The main causes of no IVT were ictus minor (28.6%) and out-of-window timeframes (22.5%), without considering sex-specific results about the potential mechanisms of these differences. However, it is observed that women have a higher percentage of exclusion due to the time elapsed since symptom onset (26.0% vs. 19.7%; *p* = 0.144), while in men, there is a higher percentage of previous excluding comorbidities (17.1% vs. 13.0%; *p* = 0.371). A total of 39.0% of patients referred to thrombectomy were dismissed from the receiving hospital in spite of the involvement of the tele-stroke center.

Female sex was associated with a higher risk of functional dependency at year 1 after an adjustment by age, secondary cardiovascular prevention, IVT, and EVT (HR 1.49; CI95% 1.37–1.61). Significant differences were observed between men and women in the progressive loss of autonomy in basic activities of daily living (measured using the Barthel score): before the episode (94.28 vs. 88.74; *p* = 0.013), one year after the episode (79.26% vs. 60.4%; *p* = 0.023), and overall (91.72% vs. 67.09%; *p* < 0.001). Overall, a higher proportion of women suffered from requiring a greater level of dependence (Barthel < 60 score) and more institutionalization (17.4% vs. 5.7%; *p* = 0.001) than men at 90 days. At one year post-stroke, despite there being parity in the total percentage of women versus men (49.1% vs. 50.9%) who had applied for assistance according to their degree of dependency, the percentage of unevaluated women is significantly lower (*p* = 0.008) than that of men (39.7% vs. 60.3%) (Figure 1).

Mortality rates during follow-up were higher (HR 2.18, 95% IC 1.18–4.04) among men in the age group of 65–79 years, but at ≥80 years of age, the differences disappear and the cumulative survival is similar (*p*= 0.385). After adjustment by age, secondary cardiovascular prevention, IVT and EVT, the overall survival hazard ratio [HR] (0.87 [CI95%, 0.85–0.88]) was higher in women than men, especially <80 years of age (*p* < 0.001). Incorporating each population’s life expectancy at the age of 70 years, and using a life expectancy of 78.2 years for men and 84.5 for women, the adjusted rate of potential years of life lost is significantly higher among men (mean 12.02, CI95% 10.7–13.3) compared to women (mean 6.54, CI95% 5.6–7.4).

### 3.2. Average Costs (Table 2)

Table 2 presents the estimated costs in the first year post-stroke, revealing patterns of resource utilization and direct medical expenses between male and female stroke patients.

Women exhibited significantly higher costs related to hypertension, the use of conventional ambulance transport, transient ischemic strokes, institutionalization, and care associated with support needs.

In contrast, they showed significantly lower costs for secondary cardiovascular prevention, non-transient stroke episodes, medical ambulances, undergoing EVT, and post-hospital discharge rehabilitation sessions. The total unadjusted costs averaged 22,605.66 (CI95% 20,442.8–24,768.4), being higher in women compared to men [*p* = 0.027].

The primary cost concept was hospital treatment (38.8%), followed by the costs associated with dependence and support needs (36.3%). If the costs associated with institutionalization were included in the dependence costs, the subtotal related to the concept of care needs would represent the highest percentage (49.1%).

Figure 2 shows the distribution of the total costs.

The outcomes were most sensitive to the comprehensive costs associated with the support needs, both from formal and informal caregivers. This suggests that the distribution of modified mRs scores at 90 days post-stroke, reflecting the level of dependence, was the most influential variable in our economic model.

## 4. Discussion

Among stroke survivors, women had a worse functional status before stroke, were older, had more severe strokes, and had a greater likelihood of post-stroke dependence. However, there were no significant differences in terms of stroke treatment, but female patients exhibited similar covariate-adjusted survival rates compared to male patients, which is consistent with the broader health–survival paradox where women tend to live longer in poorer health compared to men.

The main study’s objective was to provide evidence regarding patterns of resource utilization and health costs related to these differences between men and women after a stroke [7]. The territorial socioeconomic index [12] ranges from 86 to 90.3 in relation to Catalonia (=100%). In women, social factors such as education, gender roles, social support, integration into society, education, and emergent cardiovascular risk factors [4] have an influence on stroke risk and outcomes, as they directly influence access to preventative health care as well as acute treatment. Moreover, older women are generally more prone to physical disability than men, may experience greater socioeconomic disadvantage, and are typically more affected by changes in social networks and support.

The results have indicated that the overall impact and cost of cerebrovascular disease are higher in women than in men. These disparities may be related to clinical variations, the frequency of procedures, administered treatments, and observed outcome differences between male and female stroke patients.

The main differences indicated in previous results [20] primarily relate to three points: the payment method for a package of services according to the complexity of the hospital, the low performance of ETV, and the low incidence of recording activities of rehabilitation after hospital discharge. Furthermore, the specific details of payment rates defined according to a hospital’s level of complexity within the public health system and the hospital’s role in stroke care, as assessed at episode stage or hospital discharge, contribute to the understanding of these discrepancies.

Based on previous findings in the region, morbidity indicators revealed a higher prevalence among women for cardiovascular risk factors such as hypertension, dyslipidemia, and obesity [4,25]. While costs for specific treatments for atrial fibrillation (AF) and hypertension (HTA) patients are relatively low, estimates of the medical costs have a broad range and are mostly driven by the cardiovascular consequences and/or comorbidities [20]. Additionally, women exhibited higher rates of atrial fibrillation with poorer control of anticoagulant therapy [26], and a stroke incidence ≥ 80 years resulted in increased residual disability and institutionalization compared to men [24]. This fact may be associated with an increase in the costs associated with cardiovascular secondary prevention. The interplay between these factors is not yet fully understood, and further research is needed to better identify the causes.

The percentage of women with an mRs < 2 pre-stroke score was significantly lower (*p* = 0.014), which constitutes a limiting criterion to be considered for intravenous thrombolysis and/or mechanical thrombectomy. In addition, focusing on patients with stroke code activation can provide valuable insights into optimizing stroke care pathways, resource allocation, and healthcare utilization. However, patients without stroke code activation may present with milder or more ambiguous symptoms, leading to delayed recognition and the potential omission of thrombolytic therapy or endovascular procedures. Consequently, this could result in poorer outcomes, such as impaired functional recovery and increased long-term disability. Stroke centers without EVT capabilities, such as tele-stroke centers and primary stroke centers, are the first point of contact in acute stroke care for patients who are not within the referral area of thrombectomy-capable centers. The previous functional situation of patients and the organization and level of care among local stroke centers may affect the Door-In Door-Out time [27] and treatment decisions. In this way, the percentage of patients not meeting the necessary criteria for stroke code activation was significantly lower than the average in Catalonia (5.6% vs. 14.2%, *p* < 0.001), as was the percentage of patients in out-of-window timeframes (22.6% vs. 41%, *p* < 0.001). Since the activation cost is included in the cost of the visit to the emergency department of the referring hospital, as well as in its care protocol, there is no incentive to activate it before arriving at the receiving hospital.

Women are less likely to call an ambulance and more likely to have an unknown time of symptom onset, even when adjusted for age and socioeconomic status [28]. It may long delay the **door-to-needle time,** a priority quality factor for treatment with IVT and/or EVT [29,30,31]. It should be noted that emergency ambulance teams use the Rapid Arterial Occlusion Evaluation (RACE) scale [32,33] for assessing the severity of symptoms and identifying cases with suspected large vessel occlusion candidates for endovascular treatment with a sensitivity of 84% and a specificity of 60%. However, 39.0% of cases transferred for treatment with EVT were dismissed by the receptor hospital capable of performing ETV [21]. The underlying factors for this fact are unknown, and further research is needed to better identify them.

With regard to the **imaging cost,** current and future trends will be the use of the direct-to-angiography suite (DTAS) as the dominant strategy because not only does it improve clinical outcomes but also decreases the costs compared with the standard Transcranial Doppler (TCD) [34,35]. This option has not been included in the present study due to its current unavailability in the stroke care protocol.

Overall, the results with **IVT are consistent** [36,37,38], with evidence suggesting increased survival after IVT in women compared to men, despite more severe strokes, older age, and a higher prevalence of AF and hypertension, but they had 1.49-fold higher odds of poor functional outcome at 1 year post-stroke compared with men, independent of age.

The size of the sample treated with **EVT did not allow** the identification of differences in outcomes by sex. However, the overall results from the CICat demonstrate the undeniable benefits of EVT treatment [14]. EVT improved patient outcomes, and via economic evaluations, it was found to provide good value for money in all the randomized clinical trials evaluated [18,39,40,41,42], results that remained robust regardless of the stroke severity or age. The benefits of EVT are substantial: for every 100 patients treated, 38 have an outcome of a lower level of disability than with best medical management, and 20 more achieve functional independence (mRS 0-2). Cost savings to the Spanish NHS ranged from EUR 16,583 to EUR 44,289 depending on the patient cohort and long-term costs [42]. In recent times, ETV for acute stroke management has become the standard of care, and the cost of inaction should not exceed the cost of action in the years to come.

It is clear that the stroke patient profile in the area, characterized as women (58.8%) ≥ 80 years (73.1%) and ≥4 chronic comorbidities (94.9%), may influence decisions and costs associated with EVT despite its associated benefits [14,42,43,44]. In addition, the provision of EVT could result in a suboptimal outcome in stroke treatment for women when it is associated with other factors such as socioeconomics and accessibility [11,12,13]. Prior studies [39,45,46,47,48,49] assessing sex differences in thrombectomy outcomes found no differences in the rates of reperfusion and time to reperfusion, nor significant differences in 90-day neurological outcomes between transportation to a local stroke center (drip-and-ship) and a thrombectomy-capable (mothership model) referral center [50,51]. According to the results, women with strokes from the area would be a target population with high-cost benefits associated with residual disability and the need for post-episode care. The characteristics of the area facilitate low exclusion due to out-of-window timeframes, with greater accessibility to IVT. However, the results showed a high prevalence of cases dismissed for thrombectomy despite having a tele-stroke system operating between the originating hospital and the receiving hospital. The provision of intricate and cross-cutting care through ambulance or air transport, along with a skilled workforce in emergency medicine, neurology, radiology, and neuro-intervention, can sometimes be constrained, potentially hindering access to EVT for all eligible patients, and require further evaluation, not only for possible disparities by sex but also for inequities related to accessibility. Previous results have shown that women receive EVT less frequently than men [52,53] in spite of improved outcomes and survival. The reasons for these differences are unclear, although they might be due in part to sex-related differences in clinical presentation, the timing of presentation, differences in patients’ knowledge of stroke and response to symptoms, and physician perception of patients. Not distributing value in healthcare to regional health systems can produce socio-sanitary boomerang effects resulting from an increase in the dependent, unhealthy population.

Older women have a higher likelihood of **functional dependence** and higher mRS scores at baseline, which may have precluded them from inclusion in some of these studies [21,46,47]. Women are institutionalized more often after stroke and have a poorer recovery, regardless of study location and time period, while controlling for pre-stroke function and stroke severity [24,54]. Despite the significant differences in the assessment of stroke severity and dependency, a low percentage of patients received rehabilitation after hospital discharge, with no differences based on gender but higher costs among men. Considering that stroke is the leading cause of acquired disability, these results are clearly unexpected in relation to the published comprehensive rehabilitation protocol, even if there were issues in the activity recording [21]. The transportation to the RHB center was more expensive than the total cost of the RHB sessions. Currently, it seems evident that costs may be conditioned by clinical and geographical characteristics, and we do not know enough about their relationship from the perspective of sex. No temporal adjustment was made, given that the units of resources used correspond to the first year of the stroke episode. Furthermore, the results for the disability variable are especially sensitive to EVT accessibility, as it can modify the effects on individuals and the social costs of dependency. Therefore, there are not enough cases to conduct a viable analysis.

Rather than tailoring policies that favor specific subgroups, it would be prudent for acute stroke treatment guidelines and quality measures to prioritize increased access to EVT for all eligible patients. In particular, there is a pressing need for policies aimed at enhancing stroke recognition and facilitating accessibility to comprehensive stroke centers that offer EVT based on the degree and certainty of cost-effectiveness.

### Strengths and Limitations

To the best of our knowledge, this is the only study that has conducted a cost comparison for patients admitted to a hospital within an isochrone according to sex and gender variables. This project used the Consolidated Health Economic Evaluation Reporting Standards (CHEERS) statement [55], which recommends the minimum amount of information required for the reporting of published health economic evaluations. In addition, including a table detailing the cost per cycle [20] facilitates model replication alongside the treatment pathways.

In terms of study limitations, our analyses are constrained by the use of an observational retrospective dataset. Unknown confounding factors, such as the quality of stroke care or procedural techniques, may influence the variability in outcome scores. While acknowledging the importance of gender in shaping health outcomes, the limitations in data collection and standardization make it difficult to effectively incorporate gender as a variable in stroke research. Instead, efforts have been made to address sex-specific differences and disparities in stroke prevention, treatment, and rehabilitation based on biological characteristics. Additionally, the findings might be restricted by the specific population selected, such as those involving dismissed EVT cases, and by focusing only on patients with stroke code activation, which could lead to a misinterpretation of potential gender disparities in treatment and overall costs. Longer time horizons may also be harder to replicate because in the cost calculation, the payment methods included (by activities or by processes) and the use of a single cost for the same process or service, regardless of gender, complexity, or severity, can affect the comparability of the results. The cost allocation system not only has different characteristics that may affect comparability with other management systems but also influences treatment decisions. For all these conditions, we have not done the adjusted cost estimation by sex, which must be considered a limitation. Notably, the inclusion of neuroimaging and IVT costs in all treated stroke episodes, the independence of hospital discharge costs from the length of stay, and the absence of different costs based on severity as assessed in hospital follow-up and/or primary care are noteworthy factors.

Looking ahead, it is recommended to tailor the dataset and virtual assistants for clinicians (e.g., generative AI technology) to capture the necessary information [56], which should improve the results by including detailed interaction effects between sex and other covariates. In addition, deeper analyses are needed to explore the underlying reasons behind observed disparities, potentially through performing subgroup analyses. Currently, clinical information datasets do not allow for investigating outcomes and disparities related to gender, with specific difficulties in older age groups, especially those aged ≥80 years [42,57,58]. This emphasis is particularly pertinent to this study’s territory, where the incidence of strokes in women ≥ 80 years old is three times higher than in men.

## 5. Conclusions

Women tended to be older, with a higher prevalence of arterial hypertension diagnosis, a lower proportion of mRs ≤ 2 pre-stroke, and lower Barthel scores. Women also had a greater stroke severity and longer hospital stays, but there were no differences regarding RHB services. After a full adjustment, women had a higher risk of functional dependency (HR 1.49, CI95% 1.37–1.61) at 90 days and 1 year post-stroke, but the percentage of women not evaluated for their degree of dependency degree is significantly lower than that of men (39.7% vs. 60.3%, respectively); women were also more frequently referred to nursing or residential services. The distribution of modified mRs scores at 90 days post-stroke, reflecting the level of dependency, was the most influential variable in our economic model. In individuals aged ≤ 80 years, the use of IVT was associated with higher 1-year survival rates in women (HR 0.98, CI95% 0.99–0.97). The total unadjusted costs at first year post-stroke averaged EUR 22,605.66 (CI95% EUR 20,442.8–24,768.4), being higher in women compared to men. The primary cost concept was hospital treatment (38.8%), followed by costs associated with dependence and support needs (36.3%). Researchers and policymakers should prioritize efforts to increase knowledge of these disparities through targeted interventions, advocacy for equitable healthcare policies, and increased awareness of the social determinants of health.

## Figures and Tables

**Figure 1 healthcare-12-01369-f001:**
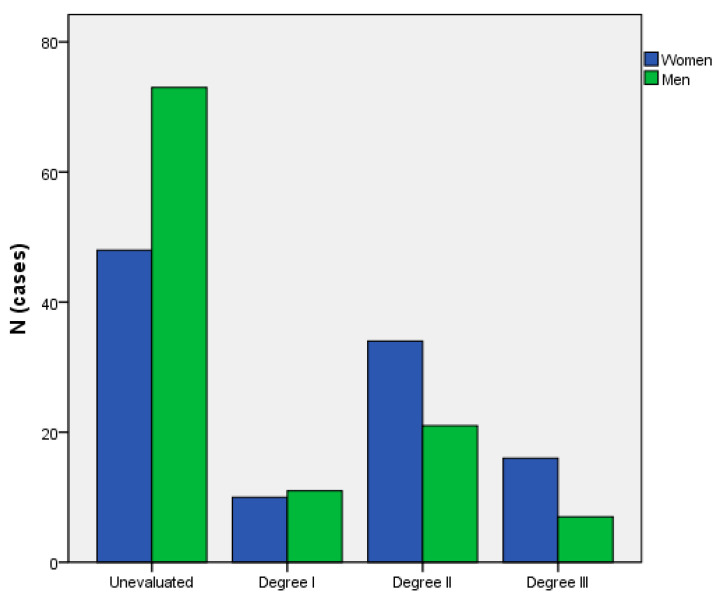
Post-stroke dependence degree (Department of Social Action and Citizenship, Government of Catalonia, Spain).

**Figure 2 healthcare-12-01369-f002:**
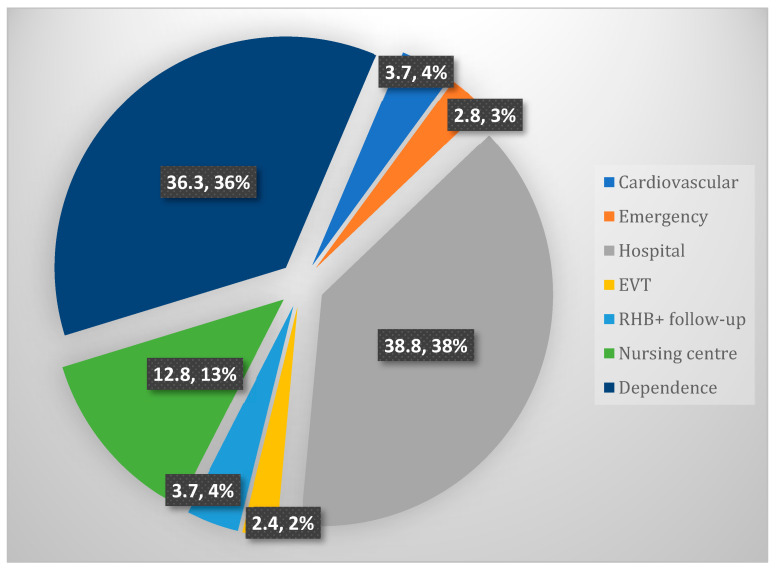
Distribution (%) of stroke costs.

**Table 1 healthcare-12-01369-t001:** Baseline description.

	All	Women	Men	*p*
	336	N_1_ = 161 (47.9%)	N_2_ = 175 (52.0%)	0.114
Age, years				
(mean ± SD)	73.3 ± 11.6	74.4 ± 11.9	72.1 ± 11.2	0.069
HTA ^1^ n (%)	222 (66.07% CI95% 60.8–71.2)	119 (73.9%)	103 (58.8%)	0.005
AF ^2^ n (%)	33 (9.9% CI95% 6.5–13.2)	16 (9.93%)	17 (9.71%)	0.908
Secondary cardiovascular prevention n (%)	69 (20.5% CI95% 16.0–25.0)	29 (18.0%)	40 (22.8%)	0.335
Ambulance	167 (88.3%CI95% 84.0–93.6)	91 (56.5%)	76 (43.4%)	0.022
Medical ambulance	20 (10.6% CI95% 5.9–15.3)	8 (4.8%)	12 (6.8%)	0.617
Air ambulance	1 (0.53% CI95% 0.01–2.9)	1 (1%)	0	NA ***
Total n (%)	188 (55.9% CI95% 50.5–61.4)	100 (62.1%)	88 (50.2%)	0.038
Pre-stroke mRs 0–2 n (%)	303 (90.1% CI95% 86.8–93.5)	138 (85.7%)	165 (94.2%)	0.014
Pre-stroke Barthel				
mean ± SD	91.7 ± 20.3	88.7 ± 24.2	94.2 ± 16.1	0.014
Emergency visits				
TIA ^3^	68 (21.7% CI95% 16.9–26.4)	36 (22.3%)	32 (18.2%)	0.429
Ischemic stroke	219 (69.9%CI95% 64.7–75.2)	104 (64.6%)	115 (65.7%)	0.921
ICH ^4^	26 (8.3% CI95% 8.3–11.5)	11 (6.8%)	15 (8.5%)	0.695
Total n (%)	313 (93.1% CI95% 90396.0)	151 (93.7%)	162 (92.5%)	0.821
Standard stroke				
n (%)				
Ischemic stroke	219 (65.17% CI95% 59.9–70.4)	104 (64.6%)	115 (65.7%)	0.921
TIA	96 (28.6% CI95% 23.6–33.5)	49 (30.4%)	47 (26.8%)	0.545
ICH	26 (7.9% CI95% 4.7–10.7)	11 (6.83%)	15 (8.83%)	0.695
SCAct ^5^				
All n (%)	52 (15.4% CI95% 11.4–19.5)	22 (13.6%)	30 (17.1%)	0.465
≤80 years n (%)	35 (18.8% CI95% 13.0–24.7)	12/78 (15.4%)	23/108 (21.3%)	0.407
Imaging (CT scan)/MRI				
n (%)	309 (91.9% CI95% 89.0–95.0)	151 (93.7%)	158 (90.5%)	0.327
NIHSS				
mean ± SD	8.8% (CI95% 7.9–9.6)	9.73 ± 6	7.83 ± 5	0.001
IVT/all stroke n (%)	18 (5.36% CI95% 2.8–8.0)	8 (5.0%)	10 (5.7%)	0.951
IVT/SCAct ^2^	18/52 (34.6% CI95% 20.7–48.5)	8/22 (36.4%)	10/30 (33.3%)	0.945
Out-of-window IVT n (%)	76 (22.6% CI95% 18.0–27.2)	42 (26.0%)	33 (18.7%)	0.144
EVT/SCAct ^2^ n (%)	9/58 (15.5% CI95% 5.3–27.7)	4 (18.8%)	5 (13.9%)	0.948
EVT/total IVT n (%)	9/18 (50% CI95% 26.0–74.0)	4 (50%)	5 (50%)	0.635
IVT + EVT/SCAct ^1^ n (%)	27/52 (52.0% CI95% 37.4–66.4)	12 (54.5%)	15 (50%)	0.965
Dismissed EVT/IVT n (%)	7/18 (39.0% CI95% 17.3–64.2)	3 (37.5%)	4 (40.0%)	0.705
No SCAct ^2^ n (%)	284 (84.5% CI95% 80.5–88.5)	139 (86.3%)	145 (82.8%)	0.465
Home discharge n (%)	241(71.7% CI95% 66.7–76.7)	113 (70.1%)	128 (73.1%)	0.631
Length of stay (days)				
Mean ± SD	8.3 ± 9.1 (CI95% 7.8–9.7)	9.5 ± 10.1	7.1± 8.2	0.017
Healthcare center pack RHB ^6^ by complexity				
Low (n)	5	2	3	
Medium (n)	11	7	4	
High (n)	7	2	5	
Total (n %)	23/293 (7.8% CI95% 4.6–11.1)	11/145 (7.58%)	12/148 (8.10%)	0.959
Home pack RHB ^6^ by complexity				
Low (n)	6	5	1	
Medium (n)	17	8	9	
High (n)	8	3	5	
Total (n %)	31/313 (9.9% CI95% 6.4–13.3)	16/150 (10.6%)	15/163 (9.2%)	0.807
Barthel score *				
Mean ± SD	66.7 (CI95% 54.3–70.4)	60.4 ± 40	79.2 ± 28	<0.001
Follow-up visits **				
PC ^7^	1680 (62.5% CI95% 60.6–64.3)	805	875	NA ***
Hospital	1008 (37.5% CI95% 35.6–39.3)	483	525	NA ***
Total/year	2688	1288	1400	NA ***
In-hospital death (n %)	45 (13.4% CI95% 9.6–17.8)	22 (13.6%)	23 (13.4%)	0.984
Nursing/residential services (n %)	38 (11.3% CI95% 7.7–14.8)	28 (17.4%)	10 (5.7%)	0.001
Dependence *				
Grade I	67 (38.7% CI95% 31.1–46.2)	32 (32.9%)	35 (45.50%)	0.128
Grade II	83 (47.9% CI95% 40.2–55.7)	48 (49.4%)	35 (45.5%)	0.707
Grade III	23 (13.3% CI95% 7.9–18.6)	16 (16.5%)	7(9.0%)	0.227
All (n %)	173/336 (51.4% CI95% 45.9–56.9)	97 (60.2%)	77 (44.0%)	0.004
Family caregiver	196 (58.3% CI95% 52.9–63.7)	97 (60.2%)	99 (56.5%)	0.567
External caregiver	12 (3.5% CI95% 1.4–5.7)	9 (5.6%)	3 (1.7%)	0.105
All (n %)	208 (61.9% CI95% 56.6–67.2)	106 (65.8%)	102 (58.2%)	0.189
Survival *	0.84 (0.80–0.87)	0.85	0.83	0.386

* 1 year post-episode; ** protocol Planifi.cat Project in primary care; *** NA: not applicable. ^1^ Hypertension (HTA); ^2^ atrial fibrillation (AF); ^3^ transient ischemic attack (TIA); ^4^ intracerebral hemorrhage (ICH); ^5^ stroke code activation (SCAct); ^6^ rehabilitation (RHB); ^7^ primary care (PC).

**Table 2 healthcare-12-01369-t002:** Stroke cost averaged by sex.

	Cost/ Unit Payment ^3^ (EUR)	All (%) Cost/Year	Women Cost/Year	Men Cost/Year	*p*
			N_1_ = 161 (47.9%)	N_2_ = 175 (52.0%)	0.114
**Cardiovascular background ^1^**					
HTA	486.08	107,909.76	57,843.52	50,066.24	
AF	4326.1	142,761.3	69,217.6	73,543.7	
Total (EUR)		250,671.06 (3.3%) (CI95% 244,496.6–256,845.4)	127,051.12	123,609.94	0.001
Secondary cardiovascular prevention	(average ± SD/patient/year)	29,636.88 (0.39%) (CI95% 28,218.3–31,053.6)	12,456.0 ± 4753.4	17,180.8 ± 3896.3	0.001
**Transport to hospital ^2,3^**					
Ambulance	532.38	88,907.46 (88.3%CI95% 84.0–93.6)	48,446.58	40,460.88	
Medical ambulance	797.84	15,956.08 (10.6% CI95% 5.9–15.3)	6382.72	9574.08	
Air ambulance	3281.3	3281.3 (0.53% CI95% 0.01–2.9)	3281.3	0	
Total (EUR)		108,145.56 (1.43%) (CI95% 107,226.7–109,063.2)	58,110.6	50,034.96	0.006
**Hospital Stage ^2,3^**					
Emergency visits					
TIA	275.0	18,700 (21.7% CI95% 16.9–26.4)	9900	8800	
Ischemic stroke	370.0	81,030 (69.9%CI95% 64.7–75.2)	38,480	42,550	
ICH	370.0	9620 (8.3% CI95% 8.3–11.5)	4070	5550	
Total (EUR)		109,350 (1.45%) (CI95% 105,220.6–113,479.3)	52,450	56,900	0.813
Standard care					
TIA	6342.7	431,303.6 (15.1% CI95% 15.0–15.1)	228,337.2 (15.9%)	202,966.4 (13.6%)	<0.001
Ischemic stroke	9636.25	2,110,338.75 (73.9% CI95% 73.8–73.9)	1,002,170 (69.9%)	1,108,168.7 (74.2%)	<0.001
ICH	12,083	314,158 (11.0% CI95% 10.9–11.0)	132,913 (9.2%)	181,245 (12.1%)	<0.001
Total (EUR)		2,925,800.35 (38.87%) (CI95% 2,841,536.7–2,870,063.2)	1,433,420.2	1,492,380.1	<0.001
EVT (EUR)	20,257.75	182,319.7 (2.42%) (CI95% 120,034.0–244,605.9)	81,031	101,288.7	0.001
**Hospital Follow-up ^1,2,3,4^**					
Hospital RHB (n %)		23/293 (7.8 CI95% 4.6–11.1)	11/145 (7.5%)	12/148 (8.1%)	0.959
Low	117.81	589.05	235.62	353.43	
Medium	161.59	1777.6	1131.2	646.4	
High	188.75	1321.25	377.5	943.75	
Sub-total (EUR)		3894.9 (CI95% 3009.3–4690.6)	2041.32	1943.58	
Home pack RHB by complexity (n %)		31/313 (9.9% CI95% 6.4–13.3)	16 (10.6%)	15 (9.2%)	
Low	242.72	1456.32	1,213.6	242.72	
Medium	627.52	10,667.78	5,020.1	5,647.68	
High	751.88	6015	2,255.6	3,759.4	
Sub-total (EUR)		18,139.14 (CI95% 18,073.6–18,206.4)	8,489.34	9,649.8	
Total RHB cost		22,124.0 (0.29%) (CI95% 18,262.5–25,987.4)	10,530.6	11,593.4	0.05
Transportation costs for RHB at the medical center (EUR).	149.8/session	51,681(0.68%) (CI95% 40,020.9–63,341.1)	24,717	26,964	NA ^7^
**Hospital discharge ^1,2,3,4^**					
Follow-up visits Hospital and primary care (EUR)	80.0/visit	215,040 (2.85%) (CI95% 75,874.7–85,405.3)	103,040	112,000	NA ^7^
Nursing/residential services ^6^ (EUR)	2124.32/month	968,689.92 (12.87%) (CI95% 884,899.5–1,052,478.4)	713,771.52	254,918.4	<0.001
**Dependence ^1,2.5^**					
Grade I	180.0	144,720 (40.1% CI95% 32.3–47.8)	69,120	75,600	
Grade II	315.9	314,233.2 (49.7% CI95% 41.8–57.5)	181,555.2	132,678	
Grade III	455.4	87,436.8 (9.6% CI95% 4.8–14.3)	87,436.8	NA	
Sub-total 6 (EUR)		546,390 (CI95% 545,570.1–547,210)	338,112	208,278	<0.001
Family caregiver		2,059,568 (94.2% CI95% 93.2–95.2)	1,019,276	1,040,292	
External caregiver		126,096 (5.7% CI95% 5.7–5.8)	94,572	31,524	
Sub-total 6 (EUR)	10,508 [20]	2,185,664 (CI95% 2,182,732.8–2,188,695.1)	1,113,848	1,071,816	<0.001
Formal care		546,390 (19,9% CI95% 19.9–20.0)	338,112 (23.2%)	208,278 (16.2%)	
Informal care		2,185,664 (80.0% CI95% 79.9–80.0)	1,113,848 (76.7%)	1,071,816 (83.7%)	
Total 6 (EUR)		2,732,054 (36.3%) (CI95% 2,729,122.8–2,734,985.1)	1,451,960	1,280,094	<0.001
Total cost (EUR)		7,595,512.3	4,068,548.1	3,526,964.2	
Average (CI95%)		22,605.66 (20,442.8–24,768.4)	25,270.4	20,154.0	0.027

^1^ Shared clinical records (HC3); ^2^ CiCat [21]; ^3^ Gazette of the Government of Catalonia; ^4^ Planifi.Cat Project; ^5^ Dpt of Social Action and Citizenship [25]; ^6^ income by month from Department of Social Action and Citizenship (formal care cost) at 1 year; ^7^ not applicable.

## Data Availability

The data presented in this study are available on request from de corresponding author.

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
