# Peer review of "Sex Disparities in the Direct Cost and Management of Stroke: A Population-Based Retrospective Study"

_healthcare, 2024, doi:10.3390/healthcare12141369_

Round 1

Reviewer 1 Report

Comments and Suggestions for Authors

Thorough work on the association of sex differences with the cost of stroke related care. The value of the publication is enhanced by the carefully analyzed strengths and limitations. The "population-based study" in the title is worth considering. Usually, this term refers to a larger and randomly selected population, rather than the analysis of a set of 336 hospitalized patients. After a well-conducted discussion, more unifying conclusions should be made by reducing their number. The text should be proofread to avoid transcribing errors (for example, ITV instead of IVT - line 310)

Reviewer 2 Report

Comments and Suggestions for Authors

Dear Authors,

The reviewed paper is a valuable research, however, it needs improvement.

Please find below some recommendations that might help to improve the manuscript.

Abstract:

I am afraid in the current state the meaning of the abstract is hard to understand. The abstract needs major revision. Suggest rewriting the abstract with respect to title and aim of the study, providing relevant background, methods, results and conclusions.

Please thoroughly check the manuscript for typos (e.g. line 108: "at least one of the . ").

Lines 50-53:

Readers will appreciate if authors support their statement by justifying reasons for sex/gender-specific data collection necessity.

Lines 55-58:

The paper might benefit if authors provide supportive citations of their statements with relevant sources specifying region, country, world.

Lines 68-70:

International readers will appreciate if authors explain in more detail their choice of the Catalonia region.

Introduction

The paper might benefit if authors deepen explanation of gender-related disparities for stroke management based on previous studies. This will help readers to understand a research problem, and the aim of the paper.

Materials and Methods

It seems that the Materials and Methods section is too wordy. Maybe to shorten the section avoiding general text information? A detailed description of those can be transferred to appendices. May be to strengthen the section by introducing any graphs, flowcharts, picture, etc. that work for better understanding of the content in the Materials and Methods section.

Line 115:

Please specify the meaning of the "Stroke Code". Do you mean the protocol for a rapid response system to significantly reduce thrombolysis timelines?

Line 127:

The paper might benefit if authors explain why they used data from 336 patients. Please justify a total number of patients in the study.

Figure 1:

Please explain in the text the differences between Degree I, II and III. Also, a brief explanation of the graph might sustain further reading.

Lines 352-380:

Stroke costs assessment is the main objective of the paper. However, the paper lack of explanation of costs assessment, calculation approaches. This might distract prospective readers from further reading of thiis valuable paper.

To strengh the paper, please introduce a subsection providing formulas and explanations on how the costs in the Table 2  were calculated (estimated).

Discussion

Although the paper is interesting to read, the discussion needs improvement to clearly indicate readers the practical application of the main findings, or their perspectives with respect to both cost and management issues. Maybe to split the discussion into two subsections, and provide point by point statements, explanations, and authors attitude.

Also, the title of the paper contains "management" topic, however the effect of gender disparities on stroke management is uncovered in the paper. Suggest providing results and their explanation (subsection) for management issue in the Results and Discussion sections. Also, providing relevant conclusions for stroke management in the Conclusion section will help readers to understand the observed effects.

 Conclusions

The conclusions need improvement with respect to connection healthcare outcomes on the one hand and cost and managerial issues, on another.

In general, the paper has a research potential and value for the readers of the journal, however it needs major revision.

Regards,

Reviewer

Reviewer 3 Report

Comments and Suggestions for Authors

The paper present the "Sex disparities in the Direct Cost and Management of Stroke: a population-based retrospective study."

Although the question is of interest for those in the field, the paper is difficult to read. The structure of the paper and english must be extensively reviewed.

Abstract

L30, L34 Some words are missing

L37 What is the currency unit?

Introduction

The goal of the study is unclear. In the abstract the authors mention that they will evaluate stroke costs differences based on gender. In the introduction, they propose to analyse and discuss association of sex differences with the cost related to stroke procedures and complexity according to some frameworks we do not know about. Elements of the previously published frameworks should be explicit in this paper.

M&M

Study design.

If some data were collected retrospectively whenever feasible, does that means that the manual collection was done prospectively? This is unclear.

Territory scope

L101 For the RSTE territory, are the 178112 individuals all adults?

L102-103 The aging index need to be defined. What is the unit?  Does the aurthers have a bibliographical referene?

L108 typo

L115 what is the Stroke Code?

L127-128. Are the  336 patients selected from the RSTE area?  Why not say so.

Dataset

L143-145 redondant with L127-129

L156 Are those data ecological or linked to the patient by patient ID?

L166 Is the HC3 linked to the CICat for each patient? Is it a perfect linkage or a probabilistic linkage?

Globally, it is unclear which data are ecological and which are individual? The authors sholud clarify this point.

L207 What is a reasonable request?

The list of variables which is difficult to read

The different groups of variables are heterogeneous in terms of details, length of explanation. L215 to 225 the mix of   “,”   “;”   “and” for the list of variables is confusing. A table that summarize the variable is quality and it’s source would be appreciated. It could be place in Appendix is that does not fit the journal’s requirement.

Statistical methods

Computing Mean and SD on ordinal (thus categorical) variables is not correct unless those variables are on a validated linear scale.

L271 “A total of 336 stroke activations were included” is that equivalent to patients or 1 patient could be registered more than once (more than 1 stroke per patient)?

Table 1 What is the basal description. Do you mean at baseline, when the patient was first hospitalized for their stroke event ? I recommand to have the “ALL” column on the left  side of the table between the variable names and the Wommen column. The variables sholud be grouped by (1) patients’ pre-stroke characteristics (2) acute phase and (3) post-stroke phase. Although, can post-stroke phase information considere baseline information ?

Figure 1 Title or caption of the figure sholud mentin the source of the Dependence degree measure. Does not the dependence depend on the age of the patient or other parameters ? The figure may be enriched with a third  dimension which might help explaining the “unevaluted”.

Figure 2 Poor quality

Table 2 Same comments as for Table 1. The table is really difficult to read. There is a mix of commas and dots for the thousands

Conclusions The conclusions are written as results. Although you might want to highlight the principal result, no p-value should appear in the conclusion. The conclusion should conclude on facts and answer you research question plus include public health advices and perspectives.

Comments on the Quality of English Language

The structure of the paper and english must be extensively reviewed.

Reviewer 4 Report

Comments and Suggestions for Authors

Abstract

Positives: The abstract succinctly summarizes the study's intent, methods, and general findings, which is commendable. It effectively sets the stage for the detailed analysis that follows. Authors should Clarify that the study is retrospective and specify the study area directly within the abstract to align with STROBE guidelines and provide immediate context for the reader. Also, authors should include exact statistical results for primary outcomes to strengthen the abstract's impact. For instance, stating the mean cost difference between men and women directly would offer a clearer summary of findings.

Introduction

The introduction effectively establishes the relevance of studying sex disparities in stroke management, highlighting its importance in the context of global health. Authors should integrate a more detailed discussion of global versus regional data on sex disparities in stroke outcomes and costs. This would enrich the background against which the study is framed. Authors should also clearly articulate the novel insights this study offers beyond the existing literature, particularly focusing on the economic analysis of sex disparities.

Methods

The detailed description of the data collection methodology, including the use of ICD-10 codes and the retrospective nature of the study, is a strong point. Authors should provide clarity on why specific hospitals were selected to enhance understanding of the representativeness of the study. Authors should also provide a detailed methodology for how costs were calculated, distinguishing between direct and indirect costs for reproducibility. Authors also need to justify the choice of statistical tests used, particularly the use of t-tests, and discuss their appropriateness given the data distribution. The STROBE criteria/checklist for reporting observation study should be adhered to

Results

The presentation of comprehensive data on stroke costs and outcomes differentiated by sex is valuable and informative. They should also improve the presentation of results by including detailed interaction effects between sex and other covariates such as age and comorbidity.  A deeper analyses to explore the underlying reasons behind observed disparities, potentially through subgroup analyses are needed.

Discussion

The discussion makes an important connection between the findings and broader healthcare issues, such as the potential for personalized medicine. There is need to expand the discussion to include a deeper examination of potential biological or societal reasons for observed disparities. Authors should further improve the  discussion on the applicability of the findings to different healthcare systems and demographic profiles to bolster the study's external validity.

Limitations

Acknowledgment of some limitations related to the study's retrospective design is a good practice. A more thorough discussion of biases inherent in the retrospective design and the implications of using electronic health records. The investigators should also address potential missing data issues and their impact on the study’s reliability.

Conclusion

The study concludes with a reflection on the significant impact of sex disparities on stroke costs and outcomes, which is crucial for policy and clinical practice. Make explicit links between the observed economic impacts and potential policy changes or clinical practice modifications. Provide specific, actionable recommendations based on the findings to help mitigate the observed disparities, such as targeted interventions or policy modifications.

Overall, this manuscript provides valuable insights into sex disparities in stroke management costs but requires several revisions to enhance its clarity, depth, and practical relevance. These changes will ensure the study not only contributes to academic discourse but also has practical implications for healthcare policy and practice.

Comments on the Quality of English Language

Minor English edits

Reviewer 5 Report

Comments and Suggestions for Authors

I have read the paper with interest. Although the study reveals some insights and empirical evidence on the subject matter, I have serious concerns on the manuscript. Main points are listed below. 

Comments:

·         The abstract section has many statements without complete sentences. It does not provide information on methods of analysis. Conclusion sentence is not related with the aim of the manuscript. Overall, the abstract has serious problems.

·         The contribution of the study is not clear. What is the place of this study in the related literature?

·         The data collection procedures are not described systematically. Are there any ethical approval processes?  What data are patient level? What data are collected at hospital level or regional level? Are these data sets combined?

·         Authors should pick up either gender or sex and use it consistently in the manuscript.

·         Number of variables and their descriptions are not clear.

·         There are many statistical methods used in the study. However, the manuscript does not explain why these methods are chosen? No justifications are provided.

·         A methodological concern for the study is the potential for omitted variables and/or control variables. Authors should explain why they do not use statistical models to control for variables in analysis.

·         Are distributions variables appropriate for the chosen methods? Any skewness?

·         Conclusion section is a list of items with numbers. This does not satisfy requirements for conclusion of scientific study.

·         The manuscript lacks the flow and style of a scientific article.

Round 2

Reviewer 2 Report

Comments and Suggestions for Authors

Dear Authors.

The authors improved the article, although not all of the reviewer's suggestions were taken into account. Nevertheless, the article in its current state is of value to the scientific community and general readers.

Best wishes,

Reviewer

Reviewer 3 Report

Comments and Suggestions for Authors

Dear authors

Thank you for your answers.

I still have some recommendations and points that need clarification.

I would recommend that :

-              you cite L137 the reference for the computation of the aging index.

-              you cite L149 Acker et al (2007)  for the definition of the Stroke Code activation system

1.        In section 2.6. Statistical, L346 you still state that “Continuous and 345 ordinal variables were characterized by either mean or standard deviation (SD)” although you answered that Barthel score, Length of stay, NIHSS and Age were not ordinal variables.

2.        Table 1 if I understood correctly your answer the baseline description was on date of June 2018. It remains unclear as some variables are link to the stroke event and after like death, dependence.

3.        What statistical model was used to compute the hazard ratio?

4.        What is a full adjustment?

5.        Why not adjust the cost estimation?

Reviewer 5 Report

Comments and Suggestions for Authors

The authors provided some revisions and explanations for the previous concerns. Although they made some improvements in the manuscript, my main and serious concerns on contribution, the style, and scientific analysis still stay. 
